# Durvalumab-Associated Pneumonitis in Patients with Locally Advanced Non-Small Cell Lung Cancer: A Real-World Population Study

Chloe Ahryung Lim [1], Sunita Ghosh [2], Hali Morrison [3], Daniel Meyers [1], Igor Stukalin [1], Marc Kerba [3], Desiree Hao [3] and Aliyah Pabani [3,*]

1. Internal Medicine Residency Program, University of Calgary, Calgary, AB T2N 4N1, Canada; chloe.lim@ahs.ca (C.A.L.); daniel.meyers@ucalgary.ca (D.M.); igor.stukalin@ucalgary.ca (I.S.)
2. Cross Cancer Institute, University of Alberta, Edmonton, AB T2S 3C3, Canada; sunita.ghosh@ualberta.ca
3. Tom Baker Cancer Center, University of Calgary, Calgary, AB T2N 4N2, Canada; hali.morrison@ahs.ca (H.M.); marc.kerba@ahs.ca (M.K.); desiree.hao@ahs.ca (D.H.)
* Correspondence: apabani1@jhmi.edu

**Abstract:** The PACIFIC trial led to a new standard of care for patients with locally advanced lung cancer, but real-world practice has demonstrated that immune checkpoint inhibitor (ICI) pneumonitis can lead to significant clinical complications. This study aimed to examine the clinical predictors, outcomes, and healthcare utilization data in patients who received consolidation durvalumab. Using the Alberta Immunotherapy Database, NSCLC patients who received durvalumab in Alberta, Canada, from January 2018 to December 2021 were retrospectively evaluated. We examined incidence and predictive values of severe pneumonitis, with overall survival (OS) and time-to-treatment failure (TTF) using exploratory multivariate analyses. Of 189 patients, 91% were ECOG 0–1 and 85% had a partial response from chemoradiation prior to durvalumab. Median TTF and OS were not reached; 1-year OS was 82%. An amount of 26% developed any grade of pneumonitis; 9% had ≥grade 3 pneumonitis. Male gender and a pre-existing autoimmune condition were associated with severe pneumonitis. V20 was associated with any grade of pneumonitis. Pneumonitis development was found to be an independent risk factor for worse OS ($p = 0.038$) and TTF ($p = 0.007$). Our results suggest clinical and dosimetric predictive factors of durvalumab-associated pneumonitis. These results affirm the importance of careful patient selection for safe completion of consolidation durvalumab in real-world LA-NSCLC population.

**Keywords:** durvalumab-associated pneumonitis; immunotherapy toxicity; radiation pneumonitis



## 1. Introduction

The PACIFIC study of consolidative durvalumab immunotherapy as post-curative intent concurrent chemoradiation therapy was the first study to show a significant improvement in overall survival (OS) for patients with unresectable, locally advanced non-small cell lung cancer (LA-NSCLC) [1]. Durvalumab is a selective human IgG1 monoclonal antibody that blocks PD-L1. This blocking interrupts the binding of PD-1 (expressed on T cell) and PD-L1 (expressed on tumour cells), leading to active T cells against cancer cells. However, inadvertent off-target toxicity can occur from blocking these natural checkpoints, termed immune-related adverse events (irAEs). Immune checkpoint inhibitor (ICI) pneumonitis is one such irAE that is of particular concern, due to its potential for severe and life-threatening consequences. [2]. Although the study of consolidation durvalumab became practice changing for this patient population, ICI pneumonitis remains a significant concern, potentially leading to severe clinical consequences and discontinuation of therapy.

The PACIFIC trial reported a rate of 12.6% of any grade of pneumonitis and 2.7% of grade 3 or higher pneumonitis. Pneumonitis was also reported as the most frequent

adverse event leading to discontinuation of durvalumab with 4.8% of patients in the durvalumab group [1]. Currently, the European Society of Medical Oncology (ESMO) Clinical Practice Guidelines and American Society of Clinical Oncology (ASCO) Clinical Practice Guidelines recommend first-line steroid treatment for most patients with grade 2 or higher ICI pneumonitis [3,4]. The impact of corticosteroids on the antitumor effect of immunotherapy remains unclear with some real-world studies showing worse OS and disease control in patients that have received corticosteroids, particularly in the early period after initiating ICI [5]. Further, clinical practice guidelines recommend that patients with grade 3 or higher pneumonitis should not be rechallenged with ICI and that rechallenge in other patients can be considered on an individual basis with close monitoring upon complete resolution of symptoms [3,4].

Many questions remain unanswered regarding ICI pneumonitis in the real-world setting. The incidence, clinical predictive factors, optimal management, and impact on healthcare utilization remain unknown. In this study, we sought to evaluate the real-world rates of ICI pneumonitis in a cohort of patients with LA-NSCLC treated with consolidation durvalumab as well as its impact on clinical outcomes and health care utilization. Furthermore, we explored potential risk factors for the development of ICI pneumonitis including clinical characteristics and radiotherapy dose–volume metrics.

## 2. Materials and Methods

### 2.1. Study Design and Population

We conducted a retrospective cohort study of patients of patients with a pathologically confirmed diagnosis of LA-NSCLC treated with definitive chemoradiation therapy followed by at least one cycle of durvalumab from two tertiary cancer centres of Alberta—the Tom Baker Cancer Centre in Calgary, Alberta, and the Cross Cancer Institute in Edmonton, Alberta, between 1 January 2018 and 31 December 2021. Data were primarily extracted from the Glans-Look Database, a comprehensive provincial database of lung cancer patients, and supplemented with the Alberta Immunotherapy Database, a database of patients treated with immunotherapy in Alberta. The electronic medical record (EMR) was reviewed to collect patient demographics, clinical, serological, and pathological data, treatment data, and outcome characteristics. Radiation planning dosimetric parameters were extracted from the radiotherapy treatment planning system, Aria RO (Varian Medical Systems, Inc., Palo Alto, CA, USA). Staging was based on the American Joint Committee on Cancer (AJCC) 8th Edition system.

### 2.2. Definition and Grading of Pneumonitis

Pneumonitis cases following at least one cycle of durvalumab were identified using oncologists' clinical assessments. Pneumonitis grade was based on oncologists' clinic notes and radiographical changes identified on the imaging report when available, using Common Terminology Criteria for Adverse Events version 5.0 (CTCAE v5.0) (grade 2: symptomatic, medical intervention indicated, limiting instrumental activities of daily living (ADL); grade 3: severe symptoms, limiting self-care ADL, oxygen indicated; grade 4: life-threatening respiratory compromise, oxygen indicated; grade 5: death). Pneumonitis cases occurring prior to initiation of durvalumab were excluded. Due to challenges with distinction between radiation-induced pneumonitis and ICI pneumonitis, the cases that were explicitly identified with the oncologist's notes as radiation pneumonitis were excluded (e.g., those within the radiation field, unilateral and/or single foci). Any diagnostically uncertain cases with possible mixed etiology as per the medical oncologist's notes were included.

### 2.3. Outcomes

Outcomes of interest were compared between patients with and without ICI pneumonitis. The primary outcome of this study was the development of pneumonitis. Secondary endpoints were OS and time-to-treatment failure (TTF). OS was defined as the time from

durvalumab start until death of any cause or last follow-up date. TTF was determined as the time from durvalumab start until discontinuation of durvalumab for any reason, including disease progression, treatment toxicity, and death. TTF was chosen as an endpoint rather than PFS as this is a more patient-focused outcome and allows us to address discontinuation due to treatment toxicity. Best response post-durvalumab was defined from the subsequent radiological reports during and after completion of durvalumab, if applicable, based on RECIST, version 1.1 assessments. To evaluate the impact of ICI pneumonitis on health care service utilization, we compared rates of presentation to the ED and admissions to hospital at any time during ICI treatment. ICI treatment was defined as the time from the date of first dose to a period 150 days after the receipt of the final dose. We also assessed the rate of in-hospital mortality during the first admission after initiation of ICI and at any point during ICI treatment. The Lung Immune Prognostic Index (LIPI) was used as a clinical scoring tool to stratify patients in "poor", "intermediate" and "good" prognostic groups [6].

### 2.4. Statistical Analyses

Survival analyses were performed using the Kaplan–Meier method and log-rank tests. Cox proportional hazards models were constructed to identify individual factors associated with OS and TTF. Landmark analyses with a timepoint of 3 months were completed to account for immortal time bias. Univariant and multivariant cox proportional hazards regression models were used to identify associations between patient demographics (age, sex, prior autoimmune disease, and smoking status), radiotherapy dose–volume metrics, and outcomes with pneumonitis. $p$ values less than 0.05 were considered statistically significant for our analysis. Statistical analyses were performed with SPSS software (BM Corp. Released 2021. IBM SPSS Statistics for Windows, Version 28.0. Armonk, NY, USA: IBM Corp.).

### 3. Results

*3.1. Patient Characteristics*

A total of 189 patients were identified from January 2018 to December 2021 as having unresectable LA-NSCLC treated with definitive chemoradiation followed by a minimum of one cycle of consolidative durvalumab. In the landmark analyses with a timepoint of 3 months, one case was excluded in the pneumonitis group. The median age was 67 (range 30–84 years) with 101 females, comprising 53% of our population. Most patients (91%; $n = 172$) had an Eastern Cooperative Oncology Group (ECOG) performance status of 0–1. Fifty-five percent of patients ($n = 105$) had stage IIIA at diagnosis. Over half of patients had adenocarcinoma (56%; $n = 106$), followed by squamous cell ($n = 70$), and other ($n = 12$), which included mixed squamous with neuroendocrine tumor, lymphoepithelioma-like pathology, and pleomorphic carcinoma). The majority of the patients were either current or ex-smokers (93%; $n = 177$) and the median smoking history was 40 pack years. See Table 1 for all patient characteristics.

**Table 1.** Demographics and clinical characteristics.

| Demographic | Total ($n = 189$) | Pneumonitis ($n = 49$) | No Pneumonitis ($n = 140$) | *p*-Value |
|---|---|---|---|---|
| Median age (years) | 67 (range 30–84) | 67 (range 31–81) | 67 (range 30–84) | 0.999 |
| Sex | | | | 0.546 |
|     Male | 88 (47%) | 21 (43%) | 67 (48%) | |
|     Female | 101 (53%) | 28 (57%) | 73 (52%) | |
| ECOG | | | | 0.861 |
|     ECOG 0 | 40 (21%) | 11 (22%) | 29 (21%) | |
|     ECOG 1 | 132 (70%) | 35 (71%) | 97 (69%) | |
|     ECOG 2–3 | 17 (9%) | 3 (6%) | 13 (9%) | |

**Table 1.** *Cont.*

| Demographic | Total (*n* = 189) | Pneumonitis (*n* = 49) | No Pneumonitis (*n* = 140) | *p*-Value |
|---|---|---|---|---|
| Histology | | | | 0.087 |
|     Adenocarcinoma | 106 (56%) | 34 (69%) | 72 (51%) | |
|     Squamous cell | 70 (37%) | 14 (29%) | 56 (40%) | |
|     Others * | 13 (7%) | 5 (10%) | 12 (9%) | |
| Tumour characteristics | | | | 0.331 |
|     PDL-1 ≥ 50% | 59 (31%) | 15 (31%) | 44 (31%) | |
|     PDL-1 1–49% | 45 (24%) | 12 (24%) | 33 (24%) | |
|     PDL-1 < 1% | 36 (19%) | 13 (27%) | 23 (16%) | |
|     Unknown | 49 (26%) | 9 (18%) | 40 (29%) | |
| Molecular characteristics | | | | 0.233 |
|     Epidermal growth factor receptor *(EGFR)* | 8 (4%) | 4 (8%) | 4 (3%) | |
|     B-raf proto-oncogene *(BRAF [V600E])* | 2 (1%) | 1 (2%) | 1 (1%) | |
|     K-Ras proto-oncogene *(KRAS)* | 24 (13%) | 6 (12%) | 18 (13%) | |
|     Anaplastic lymphoma kinase *(ALK)* fusion | 1 (1%) | 0 (0%) | 1 (1%) | |
|     ROS proto-oncogene 1 *(ROS1)* | 1 (1%) | 1 (2%) | 0 (0%) | |
| Smoker/past smoker | 177 (94%) | 45 (92%) | 132 (94%) | 0.545 |
| Obese (BMI ≥ 30) at diagnosis | 46 (24%) | 15 (31%) | 31 (22%) | 0.234 |
| Good LIPI score | 98 (52%) | 26 (53%) | 72 (51%) | 0.844 |
| Reason to discontinue durvalumab | | | | 0.103 |
|     Progression | 41 (22%) | 6 (12%) | 34 (24%) | |
|     Toxicity | 43 (23%) | 6 (12%) | 10 (7%) | |
|     Death/Poor ECOG | 7 (4%) | 4 (8%) | 7 (5%) | |
| Best Response post-durvalumab | | | | 0.108 |
|     Partial Response | 50 (26%) | 8 (16%) | 42 (30%) | |
|     Stable Disease | 104 (55%) | 28 (57%) | 76 (54%) | |
|     Progressive Disease | 30 (16%) | 11 (22%) | 19 (14%) | |

* Others includes mixed adenocarcinoma with squamous components, large cell neuroendocrine, and not specified; PDL-1 was ≥50% in 59 patients (31%) followed by between 1 and 49% (*n* = 45; 24%), and PDL-1 < 1% (*n* = 36; 19%). Eight patients had *EGFR* mutation with most common being exon 19 deletion (*n* = 6) followed by L858R (*n* = 1). Two patients had *BRAF* mutation, both V600E. *KRAS* mutation was seen in 24 cases. Only one case of *ALK* fusion was noted.

### 3.2. Treatment Regimen

Definitive chemotherapy regimens received included pemetrexed/platinum (37%, *n* = 69), vinorelbine/platinum (31%, *n* = 60), paclitaxel/platinum (25%, *n* = 47), and etoposide/platinum (6%, *n* = 11). External beam radiation was delivered using modern (Volumetric modulated Arc Therapy (VMAT)) planning techniques to a dose of 60 Gy in 30 daily fractions over 6 weeks using 6 and 10MV photons. QA planning review with radiation oncology, medical physics and dosimetry support was conducted on all lung cases. The median bilateral lung (lung-PTV) V5 (percentage of irradiated lung volume ≥ 5 Gy) and V20 (percentage of irradiated lung volume) were 67.65 (range 53.50–1096.27) and 29.90 (range 22.91–51.40). The median lung mean dose was 1094.04 cGy (range 24.27–1613.67). Most patients (*n* = 160; 85%) had partial response after chemoradiation prior to starting consolidative durvalumab.

### 3.3. Pneumonitis and Health Utilization Data

#### 3.3.1. Pneumonitis

Twenty-six percent of our cohort (*n* = 49) developed any grade of pneumonitis, whereas 9% of the total cohort (*n* = 17) had severe pneumonitis, defined as grade 3 or higher, at initial diagnosis (Table 2). Median time to pneumonitis from the start of durvalumab was 75 days or 0.20 years (range 2 to 451 days).

Corticosteroids were administered to 86% as treatment (*n* = 42), with the median dose being 50mg (converted to prednisone equivalent dosage). Median days on steroids were 62 days and mean of 93 days (range 5–868 days). Of note, three out of five cases of grade 1 pneumonitis received steroids.

**Table 2.** Pneumonitis outcomes and health utilization data.

|  | Pneumonitis Cohort (*n* = 49) |
|---|---|
| Grade of pneumonitis |  |
|     Grade 1 | 5 (10%) |
|     Grade 2 | 27 (55%) |
|     Grade 3 | 14 (29%) |
|     Grade 4 | 3 (6%) |
| Corticosteroid use | 42 (86%) |
|     Median dose (converted to prednisone equivalent, mg) | 50 |
|     Median days used (days) | 62 |
| ED visit from pneumonitis | 23 (47%) |
|     ED visit within first month of starting durvalumab | 7 (14%) |
| Hospitalization from pneumonitis | 17 (35%) |
|     Hospitalization within first month of starting durvalumab | 4 (8%) |
| Intensive care unit (ICU) admission | 2 (4%) |
| Hospice utilization | 0 (0%) |
| In-hospital death from complication of pneumonitis | 5 (13%) |
| Other immune related adverse events (irAE) | 14 (29%) |
|     Colitis | 2 (4%) |
|     Thyroiditis | 7 (14%) |
|     Dermatitis | 2 (4%) |
|     Other (arthritis and hepatitis) | 4 (8%) |

Abbreviations: ECOG, Eastern Cooperative Oncology Group; LIPI, Lung Immune Prognostic Index.

### 3.3.2. Other irAEs

Table 2 also highlights other irAEs noted including colitis, thyroiditis, dermatitis, arthritis, and hepatitis. These other irAEs developed in the pneumonitis cohort potentially present another reason to receive steroids outside from receiving them in the context of pneumonitis.

### 3.4. Rechallenge Following Pneumonitis

Thirteen patients (27%) were rechallenged. Only one patient had recurrent pneumonitis which was grade 1 and did not require steroids. This patient also had met PACIFIC and ESMO/ASCO rechallenge guidelines.

### Health Utilization

Within the pneumonitis cohort, almost half were seen at the emergency department (ED) by the ED physician (47%, *n* = 23), and 35% were admitted to hospital (*n* = 17). Median hospitalization stay was 10.5 days (range 3–90). Two people required intensive care unit admission. Ten percent of the pneumonitis cohort subsequently died from the pneumonitis in hospital (*n* = 5).

### 3.5. Cancer Responses

After starting durvalumab, 55% (*n* = 104) of patients had stable disease as best response on durvalumab, followed by 26% (*n* = 50) showing partial response, and 16% (*n* = 30) showing progressive disease. Two patients (1%) had no residual disease post definitive chemoradiation. The median amount of durvalumab cycles received was seven cycles (range 1–13 cycles), and 49% of patients received a full year of durvalumab therapy (*n* = 93). Median treatment duration on durvalumab was 62 days with a median follow-up duration of 613 days.

### 3.6. Survival Outcomes

Median OS was not reached, while OS for 1 year was 82% (76–87%), for 1.5 years was 71% (64–78%), and for 2 year was 64% (56–72%). The median TTF was not reached after 6 months for 85% (78–89%), and after 1 year for 80% (72–87%). OS for the pneumonitis sub-cohort for 1 year was 71%, for 1.5 years was 51% and for 2 years was 47% compared to the 'no pneumonitis' cohort for 1 year at 86%, 1.5 years at 78%, and 2 years at 68% (HR

1.72; 95% CI 1.03–2.86; *p* = 0.038). TTF for the pneumonitis cohort in 6 months was 65% and, for the 'no pneumonitis' cohort, in 6 months was 88% (HR 2.80; 95% CI 1.32–5.92; *p* = 0.007) (Figures 1 and 2). Rates of recurrence were 34% (*n* = 65) with a local recurrence rate of 13% (*n* = 24) followed by 26% (*n* = 50) of a distant recurrence rate. In the landmark analyses, there continued to be a statistically significant difference in outcomes between the pneumonitis versus no-pneumonitis cohort (Figure 3).

In multivariate analyses, male gender (HR: 4.61, 95% CI 1.08–19.75; *p* = 0.04) and a pre-existing autoimmune condition (HR: 3.99, 95% CI 0.87–18.33; *p* = 0.075) were associated with severe pneumonitis whereas V20 (HR: 1.021, 95% CI 1.00–1.04; *p* = 0.050) was associated with developing any grade of pneumonitis (Table 3). In further multivariable analysis, male gender (HR: 2.97, 95% CI 1.58–5.58; *p* = 0.001) and V20 (HR: 1.02, 95% CI 1.00–1.04; *p* = 0.007) were significantly associated with worse OS (Table 4), whereas younger age (HR: 0.95, 95% CI 0.91–1.00; *p* = 0.032), smoking status (HR: 4.19, 95% CI 1.14–15.47; *p* = 0.032), pneumonitis (HR 2.72, 95% CI 1.03–7.16; *p* = 0.043), and male gender (HR: 2.80, 95% CI 1.05–7.47; *p* = 0.039) with shorter TTF. Pneumonitis development was found to be an independent risk factor for worse OS (HR: 1.72, 95% CI 1.03–2.86; *p* = 0.038) and TTF (HR: 2.78, 95% CI 1.32–5.92; *p* = 0.007).

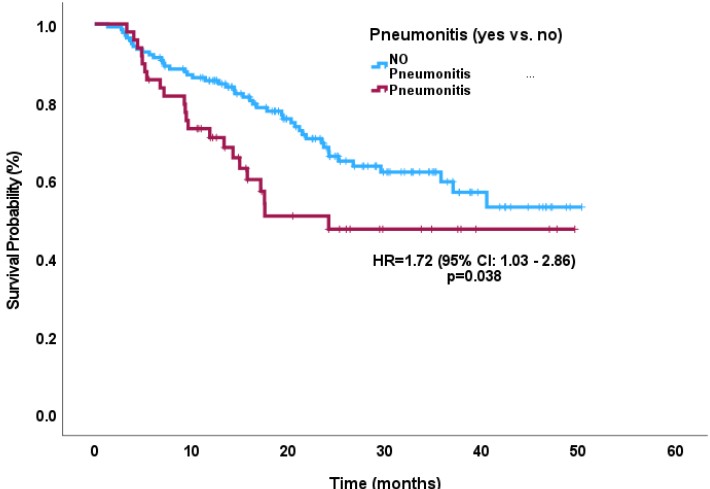

**Figure 1.** Kaplan–Meier curve for overall survival according to pneumonitis subgroup versus no-pneumonitis subgroup.

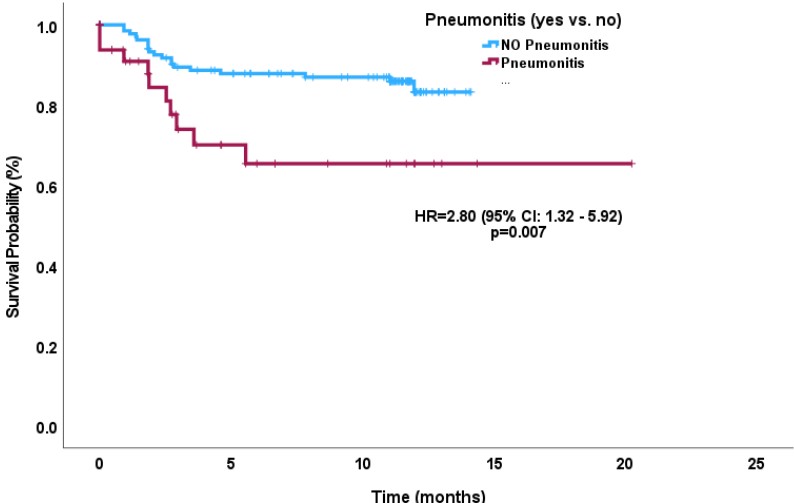

**Figure 2.** Kaplan–Meier curve for time-to-treatment failure according to pneumonitis subgroup versus no-pneumonitis subgroup.

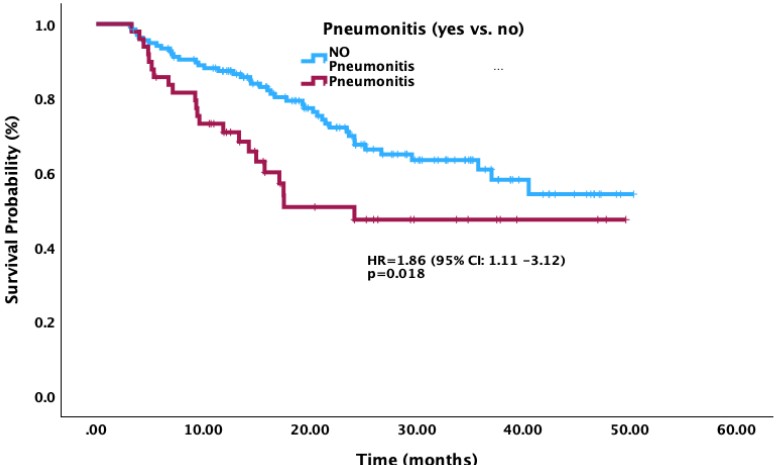

**Figure 3.** Post-landmark analyses Kaplan–Meier curves for overall survival according to pneumonitis subgroup versus no-pneumonitis subgroup.

**Table 3.** Multivariate analysis of prognostic factors for development of any grade pneumonitis.

| Variable | | Hazard Ratio | *p* |
|---|---|---|---|
| Sex | Male | 0.67 (0.30–1.52) | 0.34 |
| | Female | Reference | |
| Autoimmune condition | Yes | 1.54 (0.64–3.71) | 0.34 |
| | No | Reference | |
| Vital Statistics | Dead | 1.80 (0.79–4.10) | 0.16 |
| | Alive | Reference | |
| V20 Gy (%) | | 1.02 (1.00–1.04) | 0.05 |

**Table 4.** Multivariate analysis of prognostic factors for overall survival with durvalumab use.

| Variable | | Hazard Ratio | *p* |
|---|---|---|---|
| Age | | 0.98 (0.95–1.02) | 0.27 |
| Sex | Male | 2.97 (1.58–5.58) | 0.001 |
| | Female | Reference | |
| Autoimmune condition | Yes | 1.20 (0.59–2.42) | 0.61 |
| | No | Reference | |
| Smoking status | Yes | 1.86 (0.76–4.59) | 0.18 |
| | No | Reference | |
| Pneumonitis development | Had pneumonitis | 1.51 (0.80–2.85) | 0.20 |
| | No pneumonitis | Reference | |
| V20 Gy (%) | | 1.02 (1.01–1.04) | 0.007 |

*3.7. Next Line of Treatment*

Please see Table 5 for further details.

**Table 5.** Next line of treatment.

| Next Line of Therapy | Disease Progression (*n* = 34) | Non-Pneumonitis Toxicities (*n* = 10) | Completion of Durvalumab (*n* = 86) |
|---|---|---|---|
| Platinum-containing doublet chemotherapy | 3 | 1 | 6 |
| Single-agent chemotherapy | 3 | 0 | 0 |
| Chemoimmunotherapy | 1 | 1 | 1 |
| Immunotherapy alone (pembrolizumab) | 4 | 1 | 3 |
| Tyrosine kinase inhibitors * | 3 | 0 | 0 |
| None | 26 | 5 | 76 |
| Rechallenged | 0 | 2 | 0 |

* all tyrosine kinase inhibitors for EGFR mutant cancer.

## 4. Discussion

In this multi-centre population-based study, we report pneumonitis incidence, outcomes, risk factors, and healthcare utilization data from patients who received consolidative durvalumab in a curative setting. Compared to the PACIFIC trial (12.6%), we report a much higher rate of any grade pneumonitis (26%). Severe (grade $\geq 3$) pneumonitis was also more frequent in our cohort (9% versus 2.7%). There seems to be a large range of pneumonitis rates post durvalumab in the current literature, from 22 to 81% (Supplementary Table S1) [7–17]. A recent meta-analysis by Wang et al. also demonstrated this large range of incidences (22–48%) [18]. Comparative to other real-world studies, our cohort has lower rates of all grades of pneumonitis but higher severe (grade $\geq 3$) pneumonitis rates. We also observed a high incidence of deaths attributed to pneumonitis including five patients (3%). This contrasts with 0.8% pneumonitis-attributed deaths in the PACIFIC trial.

These pneumonitis outcome disparities between our cohort and the PACIFIC trial likely reflect the characteristics of a real-world population. This includes a subset of patients that would have been excluded in the PACIFIC trial, including those with older age, poorer performance status and, likely, patients with comorbid conditions with less physiological reserve. Furthermore, the PACIFIC trial does not describe the radiation dosimetric parameters of their patient population, limiting a direct comparison between groups and other intrathoracic characteristics of their irradiated lung cancer. We did not find a significant difference in median onset of pneumonitis between our patient population versus those on the PACIFIC trial (62 days vs. 55 days since starting durvalumab) [19]. Overall, these findings highlight that the incidence and impact of durvalumab-associated pneumonitis is potentially higher in real-world populations.

We also report the development of pneumonitis to be negatively associated with worse TTF and OS. In contrast to many other irAEs where patients generally have improved cancer outcomes, pneumonitis in NSCLC has shown conflicting data in retrospective studies, more often resulting in poorer outcomes [20–22]. This is postulated to be due to the high mortality within this patient population, commonly those with significant disease burden or baseline poor lung function [21], but this remains to be validated. Desilets et al. demonstrates that any grade pneumonitis was associated with worse OS in a cohort of 147 patients [8] compared to smaller cohort studies that did not find an association [23]. More recently, Naidoo et al. presented that severe atezolizumab-associated pneumonitis (grade $\geq 3$) across all indications was associated with worse OS [24].

Our pneumonitis cohort demonstrates high rates of patient hospitalization, presumably for hypoxia, and had multiple patient deaths, which could certainly account for worse survival within this group. Again, real-world populations often include those with poorer performance status, older age, and comorbidities including low physiologic reserve who may not be able to withstand a severe toxicity like pneumonitis. It is also possible that early discontinuation of durvalumab in patients with toxicity could potentially have contributed to the poorer outcomes from the pneumonitis cohort. Unfortunately, due to small sample sizes, we were not able to perform additional statistical testing to study this further.

To account for the frailty of the cohort with possible early deaths and immortal time bias in patients with irAEs, we conducted a landmark analysis with a time-mark of three months. This continued to demonstrate a statistically significant difference in outcomes between the pneumonitis and no-pneumonitis cohort. Further, our median follow-up time of 51 months allowed for longer follow-up, including patients who may have developed a late onset, recurrence, or chronic course of pneumonitis. Interestingly, the overall survival of our total cohort was found to be comparable to that seen in the PACIFIC trial [1]. This affirms the consistent survival benefit in the LA-NSCLC patient population but suggests caution due to the negative impact on outcomes from pneumonitis.

Prior retrospective studies reported varied results on possible predictors for pneumonitis and severe pneumonitis. In our study, V20 was associated with developing any grade of pneumonitis. This is consistent with what has previously been reported in the

literature. Gao et al. reported on several dosimetric predictors for pneumonitis including classically found V20 but also V5, V10, V40 and mean dose [25]. A smaller cohort study (*n* = 30) by Inoue et al., however, did not find severity of pneumonitis to be related to V20 [26], though Oshiro et al., with a considerably sized cohort (*n* = 91), also found V20 to be associated with ≥grade 2 pneumonitis [14]. With V20 directly influenced by the volume of disease, we speculate that V20 may modulate the biology of the lung and possibly its susceptibility of immunotherapy-related pneumonitis.

We also found that male gender and a pre-existing autoimmune condition were associated with severe pneumonitis, which has not previously been reported in durvalumab-associated pneumonitis. We speculate that male gender, due to its known predisposition to higher amounts of smoking [27], is associated with severe pneumonitis. This is also evident from our cohort's higher median smoking years in males compared to females. The relationship between underlying autoimmunity and pneumonitis has not been well-established. Aung et al. conducted a systematic review examining the safety and efficacy of ICI in patients with preexisting autoimmune disease and NSCLC, for which they reported a higher risk of grade 3 and 4 irAE in this population in pooled analysis [28]. Our results further support the careful selection of the patient population when offering durvalumab, balancing benefit versus risk for patients that may be at higher risk of developing a potentially life-threatening irAE such as pneumonitis. In clinical practice, this may involve a thorough patient-centered decision-making after discussing these risks and offering judicious monitoring. Ultimately, prospective trials are needed within controversial populations including those with autoimmune disease to help guide our management. An ongoing NCI-multicenter study is underway to evaluate the safety of ICIs among patients with underlying autoimmune diseases which may help address this question (NCT03816345).

Of note, a subset of our group had driver mutations including *EGFR*, *BRAF*, *KRAS*, and *ALK* fusion. Consolidation durvalumab appears to be less efficacious among this group of patients [29]. During the period of this study, we were not routinely performing molecular profiling to evaluate for these. We also note that the numbers are too small to extrapolate if any of the driver mutations were associated with our outcome measures.

The PACIFIC trial suggests the rechallenging option to the cohort that developed ≤ grade 2 pneumonitis after the initiation of durvalumab, which resolved to grade 1, and who achieved reduction in prednisone or an equivalent to a dose of ≤10 mg/day. We report overall successful rechallenge with only one out of 13 rechallenged reported recurrent pneumonitis and subsequent permanent discontinuation of durvalumab. Further, the recurrent pneumonitis is low grade and did not require any further steroids. There were no new irAEs reported post durvalumab-associated pneumonitis; however, 29% of patients that developed pneumonitis had other irAEs. These were most commonly endocrine in nature with thyroid dysfunction being the most common. Two small cohort studies suggested a pneumonitis recurrence rate from 28 to 33% from durvalumab rechallenge [12,30]. In contrast, Hassanzadeh et al. demonstrates similar results as our study with a small cohort of LA-NSCLC patients that developed pneumonitis post durvalumab administration of which one out of seven patients rechallenged had recurrence [23]. Saito et al., with the largest cohort of 225 patients that received durvalumab, pneumonitis relapse was only seen in three out of twenty-one patients [7]. In general, pneumonitis can be a severe and fatal toxicity, and, as such, a decision regarding rechallenge must be made with caution and close monitoring.

To our knowledge, our study is the first retrospective study with novel healthcare utilization data on durvalumab-associated pneumonitis. Previous retrospective studies have commented on treatment modality such as usage of corticosteroids and/or home oxygen therapy [7,8]. We show a considerably higher usage of corticosteroids in all grade pneumonitis than prior retrospective studies (17–23%) [7,8] but similar to smaller cohort studies (up to 100%) [23]. Specifically, we note that, in our cohort, steroids were used in grade 1 pneumonitis contrary to ESMO and ASCO guidelines [3,4], suggestive either of underlying discomfort with ICI pneumonitis leading to more lenient steroid prescription,

challenges with grading, or information bias. This higher rate is important to note as steroids may potentially dampen the antitumor immunity as seen in retrospective studies of patients with NSCLC mostly in the early course of treatment, demonstrating a lower disease control rate and poorer outcomes [31–33]. Subsequently, this can contribute to poorer outcomes including poor OS in our pneumonitis cohort with the frequent steroid use.

In our study, we found high rates of presentation to the ED for ICI pneumonitis; almost half of patients (47%) had such presentation. Many of these were subsequently admitted to hospital (35%). There are retrospective studies that show higher amounts of ED visits, hospitalization, and critical care admissions following ICI pneumonitis [34] but there are limited data specifically on durvalumab-associated pneumonitis. Our study highlights the significant utilization of healthcare resources, including ED visits, hospital admissions, and ICU admissions (4%), associated with pneumonitis. Coupled with worsened survival and increased morbidity in the pneumonitis subgroup, these findings warrant further attention for careful patient selection, patient and provider education, and vigilance of monitoring of pneumonitis in at-risk patients.

This study is limited by its retrospective nature, leading to unmeasured confounders. Second, some cases had missing documentation and an incomplete dataset due to the limitations from our healthcare electronic medical records leading to information bias. This includes information on driver oncogene mutation due to practice patterns at the time of this study. An important limitation is the attribution of cause for pneumonitis due to the considerable overlap between radiation pneumonitis and immune related pneumonitis. We attempted to control for this by excluded cases diagnosed by oncologists as solely secondary from radiation However, this remains an extremely challenging diagnosis to make with no clear radiographic or clinical features to clearly differentiate the two. Ongoing studies including machine learning to predict certain pneumonitis risks using radiologic predictors and emerging immune signatures may be beneficial for improving our understanding and prediction [35,36].

## 5. Conclusions

Overall, our provincial large retrospective study highlights the higher rates of any grade and high-grade pneumonitis in the real world compared to the PACIFIC trial. We also report a higher usage of corticosteroids for pneumonitis which may reflect local practice patterns of lenient steroid prescription or challenges with pneumonitis grading. Development of pneumonitis was associated with worse clinical outcomes, and multiple clinical and dosimetric factors were found to be associated with the development of pneumonitis and severe pneumonitis. High burden of healthcare utilization associated with pneumonitis was noted, including more ED visits, hospitalizations, and ICU admissions. As real-world populations have different characteristics that deviate from trial eligibility criteria, our study highlights the importance of careful patient selection and ongoing monitoring of patients treated with durvalumab.

**Supplementary Materials:** The following supporting information can be downloaded at: https://www.mdpi.com/article/10.3390/curroncol30120757/s1, Table S1: Literature review with large retrospective studies on durvalumab associated pneumonitis; *n* > 50).

**Author Contributions:** Conceptualization, A.P., D.H. and M.K.; methodology, A.P., D.H., M.K. and C.A.L.; formal analysis, S.G.; investigation, C.A.L.; resources, D.M. and I.S.; data curation, D.M., I.S., C.A.L. and H.M.; writing—original draft preparation, C.A.L.; writing—review and editing, D.H., M.K., A.P. and H.M.; visualization, S.G.; supervision, D.H., M.K. and A.P.; project administration, C.A.L. All authors have read and agreed to the published version of the manuscript.

**Funding:** This research received no external funding.

**Institutional Review Board Statement:** This study was conducted in accordance with the Declaration of Helsinki, and approved by the Health Research Ethics Board of Alberta (HREBA.CC-19-0380).

**Informed Consent Statement:** Patient consent was waived due to the retrospective and minimal-risk design of this study.

**Data Availability Statement:** The data presented in this study are available on request from the corresponding author.

**Acknowledgments:** We would like to acknowledge Samantha Dolter, and Heidi A.I. Grosjean for the initial sub-data collection.

**Conflicts of Interest:** The authors declare no conflict of interest.

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
