# Peer review of "Durvalumab-Associated Pneumonitis in Patients with Locally Advanced Non-Small Cell Lung Cancer: A Real-World Population Study"

_curroncol, doi:10.3390/curroncol30120757_

Round 1

Reviewer 1 Report

Comments and Suggestions for Authors

Title: Incidence, outcomes, and risk factors of acute immune check point inhibitor (ICI) pneumonitis post-chemoradiation with durvalumab for patients with locally advanced non-small cell lung cancer (LA-NSCLC): A population-based study

Summary: Authors performed a retrospective analysis on 189 LA-NSCLC patients treated with chemoradiation followed by at least one cycle of consolidative durvalumab. Authors examined predictive values of severe pneumonitis (along with several other factors) with overall survival (OS) and time-to-treatment failure (TTF) using multivariate analyses.

Authors showed that pneumonitis is independent risk factor for worse OS.

Overall, the study is a good, concise and well-designed retrospective analysis, to address the important concern of pneumonitis occurrence in lung cancer patients.

I have no comments.

Study is acceptable as it stands.

Author Response

Thanks very much for your comments!

Reviewer 2 Report

Comments and Suggestions for Authors

Dr. Chloe A Lim and colleagues studied on chemoradiotherapy for NSCLC and pneumonitis induced by durvalumab in the real-world LA-NSCLC population. Their investigation also delved into the correlation between pneumonitis development and prognosis. The paper's references appear sound, and there are no glaring issues with the understanding of pulmonary inflammation development in real-world contexts. However, certain aspects of their study—the methods, conclusions, and discussion—are unclear and render the paper unacceptable.

My comments are listed below.

Major comments:

1.        The paper lacks clarification on how they distinguished radiation pneumonitis from durvalumab-induced pneumonitis. The article suggests that "V20" is a risk factor, likely for radiation pneumonitis, and may not be relevant to durvalumab-induced pneumonitis. Despite the authors mentioning the limitation to patients who received at least one dose of durvalumab, this criterion does not effectively differentiate between drug-induced and radiation pneumonitis. This is because radiation pneumonitis doesn't manifest immediately post-radiotherapy but develops later during durvalumab administration.

2.        The authors need to provide clear insights into the factors contributing to the poor OS in patients with pneumonitis. Was the shortened OS due to the inability to administer treatment for advanced NSCLC after chemoradiotherapy owing to pneumonitis? I also wonder why the PFS results are not included. A more detailed discussion on the OS differences between the two patient groups should be conducted.

3.        Will the authors not use durvalumab in men or patients with pre-existing autoimmune conditions based on the results of this paper? It's challenging to comprehend how this paper's findings would impact daily clinical practices.

 Minor comments:

1.        Both patient groups in Table 1 are labeled "No pneumonitis." This requires correction.

2.        The definition of "Best Response post-durvalumab" lacks clarity; please provide details in the Methods section.

Author Response

Major comments:

  1. The paper lacks clarification on how they distinguished radiation pneumonitis from durvalumab-induced pneumonitis. The article suggests that "V20" is a risk factor, likely for radiation pneumonitis, and may not be relevant to durvalumab-induced pneumonitis. Despite the authors mentioning the limitation to patients who received at least one dose of durvalumab, this criterion does not effectively differentiate between drug-induced and radiation pneumonitis. This is because radiation pneumonitis doesn't manifest immediately post-radiotherapy but develops later during durvalumab administration.

Thank you for highlighting the challenges in distinguishing between radiation pneumonitis and durvalumab-induced pneumonitis. We have attempted to integrate this feedback via expanding on the methodology section via providing more clarity on our criterion: ‘Due to challenges with distinction between radiation induced pneumonitis and ICI-pneumonitis, the cases case that were explicitly identified by oncologist’s notes as radiation pneumonitis were excluded (e.g. those within the radiation field, unilateral and/or single foci). Any diagnostically uncertain cases with possible mixed etiology as per medical oncologist’s notes were included’ (line 84-88). We have also added the following statement to our limitation: ‘An important limitation is the attribution of cause for pneumonitis due to the considerable overlap between radiation pneumonitis and immune related pneumonitis. We attempted to control for this by excluded cases diagnosed by oncologists as solely secondary from radiation however this remains an extremely challenging diagnosis to make with no clear radiographic or clinical features to clearly differentiate the two’ (lines 355-360)

We agree that V20 is a well-established risk factor for radiation pneumonitis and that the association with immunotherapy related pneumonitis remains unclear. The integration of the PACIFIC study findings and applicability to Real World practice would have been enhanced by an examination of the statistical and temporal relationship between the dosimetric parameters in both the intervention and standard arms and their association to pneumonitis as described in the study.

 We would like to direct your attention to lines 281-288 where we discuss that there are conflicting data in the current literature with the V20 as a potential risk factor in durvalumab associated pneumonitis especially with the aforementioned overlap between radiation pneumonitis and durvalumab associated pneumonitis. To further elaborate, we observed similar rates of thyroiditis for instance in our population and the PACIFIC cohort. In comparison, we see a higher rate of pneumonitis in our cohort compared to the PACIFC cohort. The difference we see between these different rates of irAE may potentially be related to V20 factor which is directly related to disease volume. It is possible that V20 further modulate the biology of the lung and its susceptibility of immunotherapy related pneumonitis. We have included this in lines 288-289 for further context.

We hope that our study highlights V20 as a potential dosimetric predictor for the durvalumab associated pneumonitis including the overlap the most often see clinically.

  1. The authors need to provide clear insights into the factors contributing to the poor OS in patients with pneumonitis. Was the shortened OS due to the inability to administer treatment for advanced NSCLC after chemoradiotherapy owing to pneumonitis? I also wonder why the PFS results are not included. A more detailed discussion on the OS differences between the two patient groups should be conducted.

We would like to thank the reviewer for further clarification regarding potential reasoning of poor OS in subgroup analyses of patients that developed pneumonitis.

  1. In other literature, immunotherapy related pneumonitis, specifically high grade, are shown to be associated with worse overall survival – we have expanded it in lines 256-261: ‘In contrast, pneumonitis in NSCLC group has shown conflicting data with more often poorer outcomes seen in retrospective studies [23–25]. Naidoo et al. further presented that atezolizumab associated grade 3-4 pneumonitis across all indications were associated with worse OS [26]. The immune mediated pneumonitis and its worse outcomes is postulated due to the severity of pneumonitis on this patient population most often with significant disease burden or baseline poor lung function [24], but this remains to be validated’.
  2. In our cohort, we noted high rates of patient hospitalization, presumably for hypoxia, and even had multiple patient deaths. This can certainly attribute to worse overall survival – we included this point in lines 267-268.
  3. Another reason may be from difference in patient populations. Typically, clinical trial patient populations are healthier than real world populations due to strict inclusion criteria. In real-world practice, we also may see more patients who were previously considered borderline being treated with CRT with a hopes of deriving the hopeful benefit of consolidation durvalumab. We included the point in lines 268-271.
  4. The PACIFIC study also included eligibility criteria for radiation planning parameters of V20<35% and mean lung dose less than 20Gy. PACIFIC does not however communicate the distributions of the radiometric parameters in either the standard or intervention arm nor their statistical or temporal association with pneumonitis/radiation pneumonitis (which are grouped together) in outcome reporting.
  5. We would further like to bring your attention to the following paragraph in line 237-239 where we speculate that severity of pneumonitis with most often clinically seen significant disease burden or baseline poor lung function from smoking as a common risk factor often leading to poorer outcomes. However, there certainly can be other factors that are associated with shortened OS as highlighted by the reviewer. We have consequently amended the manuscript discussion with the following for more insight into this matter: ‘It is also possible that early discontinuation of durvalumab in patients with toxicity could potentially contribute to the poorer outcomes from the pneumonitis cohort’ (line 265-267). A spline regression can be performed to look at impact of duration of therapy on outcomes, but this was challenging to do in this patient population because of the smaller sample size.

We included the following statement to clarify why we opted to examine TTF over PFS: ‘TTF was chosen as an endpoint rather than PFS as this is a more patient-focused outcome and allows us to address discontinuation due to treatment toxicity’ (line 106-108).

  1. Will the authors not use durvalumab in men or patients with pre-existing autoimmune conditions based on the results of this paper? It's challenging to comprehend how this paper's findings would impact daily clinical practices.

Thank you to your insight on clinical implications of the study. We have modified our conclusion as following: ‘Our results further support careful selection of patient population when offering durvalumab, balancing benefit versus risk to patients that may be at higher risk of developing a potentially life-threatening irAE such as pneumonitis. In clinical practice, this may involve a thorough patient centered decision-making after discussing these risks and offering judicious monitoring when able.’  (lines 311-319).

 Minor comments:

  1. Both patient groups in Table 1 are labeled "No pneumonitis." This requires correction.

 Thank you for this observation. We have made the changes according to your suggestion.

  1. The definition of "Best Response post-durvalumab" lacks clarity; please provide details in the Methods section.

 We greatly appreciate this. We addressed this and provided details in our methodology section: line 98-100 – ‘Best response post-durvalumab was defined from the subsequent radiological reports during and after completion of durvalumab if applicable based on RECIST, version 1.1 assessments.’

Reviewer 3 Report

Comments and Suggestions for Authors

The authors of the manuscript "Incidence, outcomes, and risk factors of acute immune check-point inhibitor (ICI) pneumonitis post-chemoradiation with durvalumab for patients with locally advanced non-small cell lung cancer (LA-NSCLC): A population-based study" investigated a clinically important issue of ICI.

While I  started reading with great enthusiasm, I got disappointed. While there is a lot of important information, hard work in the study, it is difficult to follow and more information could be harvested from the data than it is presented in the manuscript. 

Some critical points:

1. The title is extremely long. Shorter title could keep the focus on the performed work. 

2. No information about durvulumab or ICI in general whatsoever in the introduction. It would help the readers focus on the importance of the clinical data analysis. (E.g. why can ICI trigger pneumotitis?).

3. Table 1 contains a lot of information but the collection of the data is selected a bit haphazardly.  Example: patients were diagnosed two main types of histology adeno- and squamous cell carcinoma, and "other". Then the list of molecular characteristics contained only the primarily adenocarcinoma drivers. Why didn't the authors provide information about the molecular studies/histology? Same for PDL1 staining? It could also provide added information about the connection to pneumonitis.

4. More focused and more structured writing style would also be recommended

Despite my criticism, I was glad to see a real-world data analysis and strongly encourage the authors to revise the manuscript.

Author Response

The authors of the manuscript "Incidence, outcomes, and risk factors of acute immune check-point inhibitor (ICI) pneumonitis post-chemoradiation with durvalumab for patients with locally advanced non-small cell lung cancer (LA-NSCLC): A population-based study" investigated a clinically important issue of ICI.

While I  started reading with great enthusiasm, I got disappointed. While there is a lot of important information, hard work in the study, it is difficult to follow and more information could be harvested from the data than it is presented in the manuscript. 

Some critical points:

  1. The title is extremely long. Shorter title could keep the focus on the performed work. 

Thank you for pointing this out. We have amended the title to following: ‘Durvalumab associated pneumonitis in patients with locally advanced non-small cell lung cancer (LA-NSCLC): a real-world population study.’

  1. No information about durvulumab or ICI in general whatsoever in the introduction. It would help the readers focus on the importance of the clinical data analysis. (E.g. why can ICI trigger pneumotitis?).

We appreciate the reviewer’s observation. To address this, we have integrated the following sentence into our introduction: ‘Durvalumab is a selective human IgG1 monoclonal antibody that blocks PD-L1. Interruption of this immune checkpoint helps T cells to recognize and destroy tumour cells. However, inadvertent off-target toxicity can occur from blocking these natural checkpoints, termed immune-related adverse events (irAE). Immune checkpoint inhibitor (ICI) pneumonitis is one such irAE that is of particular concern, due to its potential for severe and life-threatening consequences [Lee et al.]’  

  1. Table 1 contains a lot of information but the collection of the data is selected a bit haphazardly.  Example: patients were diagnosed two main types of histology adeno- and squamous cell carcinoma, and "other". Then the list of molecular characteristics contained only the primarily adenocarcinoma drivers. Why didn't the authors provide information about the molecular studies/histology? Same for PDL1 staining? It could also provide added information about the connection to pneumonitis.

Thank you for your insight. We have expanded the ‘others’ section into the following types of histology – mixed adenocarcinoma with squamous components; large cell neuroendocrine; not specified. We would also like to direct your attention to table 1 with molecular driver genes and PDL1 staining information.

To emphasize the molecular driver mutation data, we have amended the discussion with the following paragraph: ‘      Of note, a subset of our group had driver mutations including EGFR, BRAF, KRAS, and ALK fusion. Consolidation durvalumab appears to be less efficacious among this group of patients [Hellyer]. During the period of this study, we were not routinely performing molecular profiling to evaluate for these. We also note that the numbers are too small to extrapolate if any of the driver mutations were associated with our outcome measures’ (lines 304-308).

  1. More focused and more structured writing style would also be recommended

We appreciate your feedback! We have made major revisions throughout the manuscript to improve organization and readability.

Despite my criticism, I was glad to see a real-world data analysis and strongly encourage the authors to revise the manuscript.

Reviewer 4 Report

Comments and Suggestions for Authors

Article titled ‘Incidence, outcomes, and risk factors of acute immune check-2 point inhibitor (ICI) pneumonitis post-chemoradiation with 3 durvalumab for patients with locally advanced non-small cell 4 lung cancer (LA-NSCLC): A population-based study’ is interesting, because it addresses the important issue of side effects associated with immunotherapy, which is pneumonitis. I read it with interest.

Below are a few comments on the article.

1.       As for the characteristics of the group, the authors mention the EGFR, BRAF and KRAS and ALK mutations detected. It would be necessary to describe this subpopulation in more detail, since the treatment regimens could be different in first line in such a genetic change in patients with NSCLC. This should be clarified. In addition, the authors write about the ALK mutation detected, Was it a mutation or a rearrangement? This should be clarified.

2.       In Table 1, chi-square and p-values could be calculated.

3.       Next to the abbreviated gene names, full names should be included when a gene is mentioned for the first time. Please, write the gene names (full and abbreviated) in italics.

4.       Abbreviation expansions are needed: VMAT and ICU.

5.       In Table 5, for patients who have been treated with subsequent lines of TKIs, it should be specified whether molecular tests were performed for qualification for the next line, or whether they were performed before the patient was included in the first line of treatment. It should be also specify the therapeutic target of the TKI (e.g., EGFR or ALK, etc.).

6.       Citations are missing in Table 6 (in the Supplement).

Author Response

Article titled ‘Incidence, outcomes, and risk factors of acute immune check-2 point inhibitor (ICI) pneumonitis post-chemoradiation with 3 durvalumab for patients with locally advanced non-small cell 4 lung cancer (LA-NSCLC): A population-based study’ is interesting, because it addresses the important issue of side effects associated with immunotherapy, which is pneumonitis. I read it with interest.

Below are a few comments on the article.

  1. As for the characteristics of the group, the authors mention the EGFRBRAFand KRAS and ALK mutations detected. It would be necessary to describe this subpopulation in more detail, since the treatment regimens could be different in first line in such a genetic change in patients with NSCLC. This should be clarified. In addition, the authors write about the ALK mutation detected, Was it a mutation or a rearrangement? This should be clarified.

Thank you for this important observation. We absolutely agree that due to the established higher risk of pneumonitis in these molecular mutation NSCLC, especially when given TKI following immunotherapy. Most often, these patients had an identified mutation after receiving durvalumab. We unfortunately do not have all of the data available to us regarding when the mutation was identified but to mitigate this, have integrated this statement into our discussions: ‘Of note, a subset of our group had driver mutations including EGFR, BRAF, KRAS, and ALK fusion. Consolidation durvalumab appears to be less efficacious among this group of patients [31]. During the period of this study, we were not routinely performing molecular profiling to evaluate for these. We also note that the numbers are too small to extrapolate if any of the driver mutations were associated with our outcome measures’ (304-308).

We have also integrated this statement for limitations: ‘Second, some cases had missing documentation and incomplete dataset due to the limitations from our healthcare electronic medical records leading to information bias. This includes information on driver oncogene mutation, due to practice patterns at the time of this study’ (351-354).

We also thank the reviewer for their insight on specificity of ALK mutation. We clarify that this was an ALK fusion which was changed accordingly in the manuscript.

  1. In Table 1, chi-square and p-values could be calculated.

Thank you. Our manuscript has been amended accordingly.

S3.       Next to the abbreviated gene names, full names should be included when a gene is mentioned for the first time. Please, write the gene names (full and abbreviated) in italics.

Thank you for this point. Our manuscript has been amended accordingly.

  1. Abbreviation expansions are needed: VMAT and ICU.

Thank you for this point. Our manuscript has been amended accordingly.

  1. In Table 5, for patients who have been treated with subsequent lines of TKIs, it should be specified whether molecular tests were performed for qualification for the next line, or whether they were performed before the patient was included in the first line of treatment. It should be also specify the therapeutic target of the TKI (e.g., EGFR or ALK, etc.).

 We appreciate the reviewer’s insight on this matter. Please see response to #1 above. For the specific therapeutic TKI targets, we have amended table 5 to include further information.

  1. Citations are missing in Table 6 (in the Supplement).

Thank you for this point. Our manuscript has been amended accordingly.

Round 2

Reviewer 2 Report

Comments and Suggestions for Authors

I have understood the arguments made by the authors. They have appropriately revised the paper in response to my comments. There is a possibility that this paper will be accepted.

Author Response

Thanks very much for your comments!

Reviewer 3 Report

Comments and Suggestions for Authors

Dear Authors,

Thank you for your answers. 

"Durvalumab is a selective human IgG1 monoclonal antibody that blocks PD-L1". - CORRECT

"Interruption of this immune checkpoint helps T cells to recognize and destroy tumour cells". INCORRECT. In a nutshell: PD-L1 is a natural suppressor of T cell activity. If binds to PD1, which is its receptor on T cell surfaces, then T cells are inactivated and cannot destroy the tumour cell, which expresses more than the normal amount of PD-L1. If ICI-s are used, then T cells remain active causing "inadvertent off-target toxicity".

Mistakes like "patients of patients...." line 68 could be easily corrected.

Comments on the Quality of English Language

Some sentences would still need to be simplified.

Author Response

Thank you for your feedback. We have revised the manuscript as per your suggestions – “This blocking interrupts the binding of PD-1 (expressed on T cell) and PD-L1 (expressed on tumour cells) leading to active T cells.” (lines 35-36).

We have also corrected the line 68 mistake. Thank you for the recommendations!